# Structural Mutations in the Organellar Genomes of *Valeriana sambucifolia* f. *dageletiana* (Nakai. ex Maekawa) Hara Show Dynamic Gene Transfer

**DOI:** 10.3390/ijms22073770

**Published:** 2021-04-05

**Authors:** Hyoungtae Kim, Jungsung Kim

**Affiliations:** 1Institute of Agriculture Science and Technology, Chungbuk National University, Cheongju, Chungbuk 28644, Korea; rladbgus@gmail.com; 2Department of Forest Science, Chungbuk National University, Cheongju, Chungbuk 28644, Korea

**Keywords:** *Valeriana sambucifolia* f. *dageletiana*, organellar genome, gene transfer, non-plastome-originated region, variation

## Abstract

*Valeriana sambucifolia* f. *dageletiana* (Nakai. ex Maekawa) Hara is a broad-leaved valerian endemic to Ulleung Island, a noted hot spot of endemism in Korea. However, despite its widespread pharmacological use, this plant remains comparatively understudied. Plant cells generally contain two types of organellar genomes (the plastome and the mitogenome) that have undergone independent evolution, which accordingly can provide valuable information for elucidating the phylogenetic relationships and evolutionary histories of terrestrial plants. Moreover, the extensive mega-data available for plant genomes, particularly those of plastomes, can enable researchers to gain an in-depth understanding of the transfer of genes between different types of genomes. In this study, we analyzed two organellar genomes (the 155,179 bp plastome and the 1,187,459 bp mitogenome) of *V. sambucifolia* f. *dageletiana* and detected extensive changes throughout the plastome sequence, including rapid structural mutations associated with inverted repeat (IR) contraction and genetic variation. We also described features characterizing the first reported mitogenome sequence obtained for a plant in the order Dipsacales and confirmed frequent gene transfer in this mitogenome. We identified eight non-plastome-originated regions (NPRs) distributed within the plastome of this endemic plant, for six of which there were no corresponding sequences in the current nucleotide sequence databases. Indeed, one of these unidentified NPRs unexpectedly showed certain similarities to sequences from bony fish. Although this is ostensibly difficult to explain, we suggest that this surprising association may conceivably reflect the occurrence of gene transfer from a bony fish to the plastome of an ancestor of *V. sambucifolia* f. *dageletiana* mediated by either fungi or bacteria.

## 1. Introduction

Caprifoliaceae *sensu lato* is one of two large clades in the order Dipsacales, which comprises Caprifoliaceae *sensu stricto*, Diervillaceae, Dipsacaceae, Linnaeaceae, Morinaceae, Valerianaceae, and Zabelia [1]. Among these taxa, the family Valerianaceae contains approximately 350 species of annual and perennial plants, which, with the exception of Australia and New Zealand, have an extensive worldwide distribution [2]. The members of this family are characterized by sympetalous, bilaterally symmetric to strongly asymmetric flowers; inferior tricarpellate ovaries with a single carpel at maturity; a single anatropous ovule; achene fruits; the absence of endosperm; and the presence of valeriana-epoxy-triesters (valepotriates) [2,3,4]. Given the established therapeutic properties of valepotriates, a distinct class of iridoid compounds, members of Valerianaceae have been used in traditional medicine for several hundred years [4].

The genus *Valeriana* L. is the largest among those comprising the family Valerianaceae, with species distributed throughout Asia, Europe, and America [5]. Among these, *Valeriana sambucifolia* Mikan is morphologically similar to *Valeriana officinalis* L., although it can be distinguished from the latter based on the small numbers of leaflets, aboveground runners [6], and the trichomes borne on fruits [2]. *V. sambucifolia* f. *dageletiana* (Nakai. ex Maekawa) Hara (Figure 1), a broad-leaved valerian, is endemic to Ulleung Island, considered one of the hot spots of endemism in Korea [7]. Although this infra-specific taxon has for long been studied in a pharmacological context [8,9,10], there are numerous features of this plant that have yet to be sufficiently characterized. In contrast to *V. sambucifolia,* which is generally distributed in Scandinavia and Central and Atlantic Europe [11], *V. sambucifolia* f. *dageletiana* occurs only in Northeast Asia. Moreover, these plants differ with respect to the chromosome number, with that of the former and latter being 2*n* = 56 [12,13] and 2*n* = 54 or 55 [14], respectively. Despite it being taxonomically considered a forma of *V. sambucifolia*, based on the characteristics of a small number of broader leaflets (Figure 1A-B) and belowground runners (Figure 1C), we still have insufficient information about this pharmacologically attractive species [8].

Plants are characterized by two organellar genomes, namely the chloroplast genome (plastome) and the mitochondrial genome (mitogenome) [15], which in terrestrial plants have undergone independent evolution [16]. The plastomes of plants are generally conserved in terms of gene content, gene order, and genome length [17,18,19], although certain lineages, such as those of heterotrophs, have been found to have contracted genomes [20,21]. On the basis of these genomic characteristics, plastomes have been used as resources for resolving the phylogenetic relationships of uncertain lineages [22,23], providing an understanding of the transition from photosynthetic to non-photosynthetic modes of nutrition [24,25,26] and elucidating synapomorphies in the common ancestor of early diverged terrestrial plants [27,28]. In contrast to the relatively conserved plastomes, the mitogenomes of plants have undergone comparatively rapid structural evolution [29] by incorporating foreign DNA via horizontal [30] or intracellular [31,32] gene transfer. Consequently, there tends to be low similarity among the mitogenomes of terrestrial plants, even among closely related taxa [33,34,35].

Although the inter-organelle transfer of DNA from plastomes to mitogenomes has been extensively reported, transfer of DNA in the opposite direction, from mitogenomes to plastomes, appears to be a comparatively rare event that has only recently been discovered in some lineages. Moreover, whereas the former mitochondrial plastid DNAs (MTPTs) have been found to vary in length [16,36,37,38], the latter plastid mitochondrial DNAs (PTMTs) are generally smaller in length [39,40,41], although PTMTs of the *Anacardium occidentale* plastome appear to be a notable exception in this regard [42]. 

To date, the reference sequences of 4732 plastomes and 223 mitogenomes from terrestrial plants have been deposited in the NCBI database (https://www.ncbi.nlm.nih.gov/genome/ accessed on 5 April 2021). As these numbers indicate, the mitogenome sequence data are comparatively limited (only 5% compared to the plastome data), and currently there are no reference mitogenomes for plant orders such as Bruniales, Escalloniales, Paracryphiales, and Dipsacales among asterids, thereby hampering studies on mitogenome evolution and inter-organelle DNA transfer in these clades. 

In this study, we sequenced the plastome and mitogenome of the Ulleung Island endemic *V. sambucifolia* f. *dageletiana* with the following aims: (1) to compare the genetic variation between the whole plastomes of two closely related valerians, *V. officinalis* and *V. sambucifolia* f. *dageletiana*, thereby characterizing this endemic species at the genomic level; (2) to produce a reference mitogenome sequences for the order Dipsacales (to the best of our knowledge, the first reported); and (3) to gain a comprehensive insight into inter-organelle DNA transfer in the order Dipsacales.

## 2. Results

### 2.1. Variation in the V. sambucifolia f. dageletiana Plastome

The average coverage of the *V. sambucifolia* f. *dageletiana* plastome was 2037.8 ± 233.6 (mean ± SD). The sequenced plastome has a length of 155,179 bp, including 85,334 bp large single-copy (LSC) and 15,243 bp small single-copy (SSC) regions and two inverted repeats (IRs) covering 27,301 bp. The plastome contains 134 genes, encoding 82 protein-coding genes, 8 ribosomal RNAs, 39 transfer RNAs, 2 pseudogenes (*accD*, *ycf1*), and 3 partial genes (*ndhD* and 2 *trnR*-ACG in the IR region). Compared with the plastome of *V. officinalis*, the most closely related taxa in the genera, the sequenced plastome differs in that the *clpP*, *rps3*, *rps18*, and *ycf2* genes remain intact.

Notably, we found that the *trnH* gene, which, with the exception of certain *Patrinia* species, is generally located in the LSC region of Caprifoliaceae plants [43,44], has undergone translocation to the IR region in the plastome of *V. sambucifolia* f. *dageletiana* via IR expansion. Similar IR expansion and contraction at the boundaries between the IR and single-copy (SC) regions were found to account for differences in the lengths of the two *Valeriana* plastomes. With reference to the *V. officinalis* plastome structure, we assumed that IRb initially expanded to *rpl32* over *ndhF* and that, subsequently, both the *ndhF* and *rpl32* genes were eliminated from IRb and a copy of *trnR*-ACG was pseudogenized by a deletion in both IR regions. Finally, IRb expanded to within *ndhD* over *ccsA* and *trnL*. As a consequence, the IR region of *V. sambucifolia* f. *dageletiana* is now 3502 bp longer than that of *V. officinalis* (Figure 2). 

Of the 155,179 bp of the full-length *V. sambucifolia* f. *dageletiana* plastome, 10 regions, comprising 152,753 bp, were found to correspond to the plastid DNAs of other viridiplantae. However, among the unmatched sequences, we identified eight non-plastome regions (NPRs), two of which are duplicated in the IR regions, that did not correspond to any other viridiplantae plastid DNAs: one of them is located in *rbcL-accD* (LSC region), two are positioned in *trnL*-*trnL* (IR regions), and five are found in *ycf1* (SSC region). Moreover, we were unable to match six of these NPRs to any other nucleotide sequence. The 183 bp low-complexity region of NPR_4 was found to correspond to certain nuclear genome sequences or mRNAs of eukaryotes (Appendix A). Interestingly, our database search revealed that the 348 bp NPR_7 corresponded primarily with sequences in animal chromosomes, notably those of bony fishes, along with a few matches with fungal and bacterial sequences. To verify the presence of these NPRs in the *V. sambucifolia* f. *dageletiana* plastome, we re-examined the occurrence of these regions using other genomic DNA extracted from a different individual of *V. sambucifolia* f. *dageletiana* deposited in our laboratory and three individuals of *V. sambucifolia* f. *dageletiana* with different voucher specimens acquired from the Plant DNA Bank in Korea (PDBK). The eight NPRs in each of the extracted DNAs were re-sequenced using the primer sets newly designed in this study. With the exception of the number of poly-G bases in NPR_1 and a 12 bp deletion in NPR_7 in one sample, all sequenced regions were found to be identical to the initially determined plastome sequences of *V. sambucifolia* f. *dageletiana*. Additionally, the average depths of the 10 NPRs were almost identical to those of their flanking sequences. Consequently, we assumed that these NPRs are true *V. sambucifolia* f. *dageletiana* plastome sequences, as opposed to copies of genomic regions originating from the nucleus or the mitogenome.

Insertions or deletions (indels) were found to be common features of the plastomes of both *V. sambucifolia* f. *dageletiana* and *V. officinalis*. Given the difficulty in determining whether the indels in these plastomes represent insertions or deletions, based on the available data, for the purposes of the present study, we simply defined all detected indels as deletions, compared with the sequence of the other plastome. In addition to 79 deletions caused by IR expansion/contraction at IR–LSC junctions, we detected a further 205 deletions 1–762 bp in length (Figure 3A), which we classified into four types (I to IV) (Figure 3B). Type I deletions were caused by simple sequence repeats (SSRs), whereas those in categories II and III included a single pair of repetitive sequences. If there was an intervening sequence of 1 bp or no intervening sequence between two repeats, we considered these as tandem repeats and defined the deletion of one repeating unit as type II. Type III deletions were defined as those including an intervening sequence more than 2 bp in length and a repeating unit. The remaining deletions that were unassociated with repetitive sequences were classified as type IV. We established that a total of 35 type I deletions were caused by length variations in homopolymer repeats of less than 7 bp. Type II deletions were found to occur more frequently within the plastome of *V. sambucifolia* f. *dageletiana* than in that of *V. officinalis*, with those longer than 30 bp being detected exclusively in the *V. sambucifolia* f. *dageletiana* plastome. Similarly, type III deletions were detected solely in the plastome of *V. sambucifolia* f. *dageletiana*. In contrast to type II and III deletions associated with repetitive sequences and slipped strand mispairing [45], those in the type IV category were detected mainly within the *V. officinalis* plastome. 

We also identified 246 SSRs in the plastome of *V. sambucifolia* f. *dageletiana*, among which mono-nucleotide repeats accounted for the overwhelming majority of 95.2% (85% for A/T and 10.2% for C/G) (Appendix A). Although di-, tetra-, and hexa-nucleotide repeats were also identified, we were unable to detect any tri- or penta-nucleotide repeats. Consistent with these observations, at the genus level, mono-nucleotide repeats appear to be particularly prevalent throughout Caprifoliaceae (Appendix A). Interestingly, however, whereas penta-nucleotide repeats were not found in any genera, hexa-nucleotide repeats were identified in all genera within the family. 

### 2.2. Features of the V. sambucifolia f. dageletiana Mitochondrial Genome 

The mitogenome of *V. sambucifolia* f. *dageletiana* was found to be 1,187,459 bp in length, with an average coverage of 199.4 ± 32.7, a GC content of 42.5%, and 74 genes encoding 32 protein-coding genes, 5 ribosomal RNAs, 32 transfer RNAs, and 5 pseudogenes. Compared with the mitogenomes of Asterales species, the following elements, which play roles in mitochondrial electron transport and oxidative phosphorylation, remained intact: NADH dehydrogenase subunits in complex I, apocytochrome in complex III, cytochrome *c* oxidase subunits in complex IV, ATP synthase subunits in complex V, and *ccm* genes in cytochrome *c* biogenesis system I (Table 1). However, certain genes encoding ribosomal proteins (*rpl16*, *rps1*, *rps13*, and *rps19*) were identified as pseudogenes, owing to the presence of internal stop codons. Among the 32 tRNAs, 14 are similar to the plastid tRNAs of *V. sambucifolia* f. *dageletiana,* namely *trnD*-GUC, *trnF*-GAA, *trnH*-GUG (2), *trnM*-CAU (2), *trnN*-GUU, *trnP*-UGG (2), *trnS*-GGA, *trnT*-UGU, *trnV*-GAC, and *trnW*-CCA (2).

Among the 1,187,459 bp of the mitogenome, 1313 regions encompassing 749,000 bp were found to correspond to sequences in the mitogenomes of 364 terrestrial plants. However, we were unable to detect any sequences in these plants corresponding to a further 1015 regions with lengths in excess of 30 bp. Excluding the 39 regions duplicated within these 1015 non-mitogenomic regions, the remaining 976 regions cover 418,671 bp and range in length from 30 to 5965 bp, and we found that 83.4% of the sequences showed no correspondence with sequences in the searched nucleotide database, based on the search options 11-word size and 1e^-3^ expected value (e-value). To determine whether a substantial proportion of large-sized mitogenomes is unique, and if so, whether these unique genomic sequences are a consequence of nucleus-to-mitochondrion gene transfer, we examined five mitogenomes of similar lengths to the *Valeriana sambucifolia* f. *dageletiana* mitogenome, characterized by corresponding sequences, three of which were found to show correspondence to whole nuclear genome sequences (Table 2). BLAST results revealed that the proportion of regions showing no matches with sequences in the nucleotide database ranged from 12.0% (*Acer yangbiense*) to 63.5% (*Platycodon grandiflorus*), whereas portions of mitogenomes ranging from 1.1% (*Acer yangbiense*) to 25.0% (*Cucurbita pepo* subsp. *pepo*) were identified in the corresponding nuclear genomes.

We established that dispersed repeat regions account for 17.1% of the total *Valeriana sambucifolia* f. *dageletiana* mitogenome and that these regions are evenly distributed throughout the genome (Figure 4A). Moreover, we found that for repeats less than 100 bp in length, the sequence length was inversely proportional to the percentage pairwise identity between dispersed repeats. In contrast, however, we detected no significant correlation between length and pairwise identity among the repeats exceeding 200 bp in length (Figure 4B). Furthermore, compared with repeats of Asterales, the lengths of identical repeats were found to be of medium size (Appendix A). 

We also identified 1205 SSRs in the mitogenome of *V. sambucifolia* f. *dageletiana*, 901 of which are mono-nucleotide repeats, comprising 74.8% of the total (Appendix A). With the exception of tetra-nucleotide repeats, the occurrence of which was found to be more frequent than that of di-, tri-, penta-, or hexa-nucleotide repeats, the length of the motif was inversely correlated with the frequency of repeat.

### 2.3. Intercompartmental Gene Transfer in the Mitogenome of V. sambucifolia f. dageletiana

Although phylogenetic analysis would ostensibly appear an effective approach for investigating the direction of intercompartmental gene transfer, certain mutations, such as those causing rearrangements, deletions, and contig degradation, tend to prevent alignment between transferred DNAs and the sequences of origin. Compared with mitogenomes, we assumed that the acquisitions of foreign organellar DNA would be relatively uncommon in the plastomes of terrestrial plants [38,41,46,47], even though early terrestrial plant lineages, such as mosses, appear to have relatively conserved mitogenomes [48]. On the basis of this assumption, we reasoned that if DNAs prevalent in the plastomes of these plants are also found in the mitogenomes, then they should be considered to have originated from the plastomes and not vice versa.

In the present study, we detected 162 regions in the mitogenome of *V. sambucifolia* f. *dageletiana* that either partially or entirely correspond to sequences in terrestrial plant plastomes, including 13 duplicated regions. To determine the direction of gene transfer between the two organellar compartments, based on how data were categorized by certain minimum thresholds, we used three thresholds (40%, 10%, and 1%), depending on how many two-organellar genomes were hit against queries. Among 149 regions, we found that 68 corresponded to sequences in more than 40% of 9,025 plastomes and in more than 1% of a further 364 mitogenomes (Table 3 and Figure 5). These regions were accordingly considered mitochondrial plastid DNAs (MTPTs) that may have been translocated from the plastome to the mitogenome. Conversely, we identified 46 regions corresponding to sequences in at least 150 mitogenomes but to sequences in less than 1% of plastomes (Table 3 and Figure 5). These regions were thus considered to be plastid mitochondrial DNAs (PTMTs), which may have undergone translocation from mitogenomes to plastomes. However, the complexity of the sequences of other regions precluded an assumption of their origin. Only four regions of the seven that showed comparatively low correspondence to sequences in organellar genomes were similar to sequences in certain plastomes of Orobanchaceae or Caprifoliaceae.

## 3. Discussion

### 3.1. Frequent Gene Transfers from the V. sambucifolia f. dageletiana mitogenome

Although the mitogenomes of early diverged terrestrial plants, such as mosses, tend to be stable with respect to size and structure [48], these genomes have undergone comparatively rapid increases in size and structural diversity in the more evolutionarily advanced vascular plants [49,50,51]. The notable differences in the lengths of vascular plant mitogenomes are believed to be attributable to the transfer of genes from the co-occurring nucleus and chloroplasts [16,39,52] or that of foreign DNA derived from other plants [30,53] and duplications within the mitogenome [54,55,56].

With the exception of the mitogenome of *Platycodon grandifloras*, the length of the *V. sambucifolia* f. *dageletiana* mitogenome is three to four times that of the mitogenomes in other plants within Asterales, although without any extensive variation in gene content. In this regard, although dispersed repeat sequences account for 17.1% of the *V. sambucifolia* f. *dageletiana* mitogenome, the primary factor contributing to an expansion in genome size, accounting for 31.6% of the entire mitogenome, is the presence of ostensibly unique regions comprising sequences for which we obtained no matches with sequences in the mitogenomes of other vascular plants reported to date. However, far from being unique to *V. sambucifolia* f. *dageletiana*, the presence of substantial unmatched portions of the mitogenome appears to be a common phenomenon associated with large-sized plant mitogenomes. Interestingly, however, some of these unmatched sequences were found to correspond to sequences in the nuclear genome of the same organism (Table 2). 

On the basis of our understanding of the monophyly of terrestrial plants, it might be assumed that there would not have been high mitogenome sequence diversity among the populations of ancestral plants. As these plants evolved, however, there would have been increasing opportunities for bidirectional intracellular gene transfer between mitogenomes and nuclear genomes [32,46,57], along with the degradation and deletion of foreign DNAs in both nuclear and mitochondrial genomes, during the diversification of plants [16,58] (Figure 6). Accordingly, these events may have led to alterations in mitochondrial-incorporated nuclear DNA such that these sequences diverged from those of the original nuclear DNA, thereby obscuring identification of the origins of mitogenome sequences in terrestrial plants. In addition, the horizontal transfer of genes from other plant genomes to mitogenomes may have contributed to greater sequence and structural complexity [53]. Accordingly, these series of evolutionary events, involving gene transfer and decay, can be assumed to have led to a marked diversification in mitogenome sequences among the different orders, families, and genera of terrestrial plants, with the presence of certain modified sequences becoming intrinsic features of the mitogenomes of these plants. 

Unfortunately, apart from the mitogenome of *V. sambucifolia* f. *dageletiana* sequenced in the present study, there is currently no information available regarding the sequences of mitogenomes in the order Dipsacales, and to date, there have been no reports of nuclear genome sequences in species of the genus *Valeriana*. Consequently, at present, we are unable to establish the identity of those sequences transferred directly to the mitogenome of *V. sambucifolia* f. *dageletiana*. Nonetheless, it is clear that there has at least been a frequent transfer of genes into mitogenomes within the Dipsacales lineage, and further study of the nuclear genome should accordingly be undertaken to gain a more in-depth insight into the intrinsic portions of incorporated mitogenomic DNA at the species level. 

### 3.2. Origins of Non-Plastomic Regions Found in the V. sambucifolia f. dageletiana Plastome

Although in theory, six types of reciprocal inter-compartmental gene transfer among the three plant genomes are possible, the occurrence of gene flow from either the nucleus or mitochondria to plastids has been considered to be extremely infrequent [46,47], given the structural compactness of plastomes, limitation of non-homologous recombination, lack of an efficient uptake system for exogenous DNA, and the improbability of inter-chloroplast fusion [47]. However, accumulating genomic evidence is providing new insights into the intracellular transfer of genes from the mitogenome to the plastome in a number of different plant lineages: Apiales [40,59], Vitales [39], Spindales [42], Poales [60], and Gentianales [41,61]. Moreover, the horizontal transfer of genes from the genomes of distinctly related species, mainly bacteria, to the plastomes of non-plant organisms [62,63,64,65] and ferns [66,67,68] has also been reported. 

In the present study, we identified eight NPRs, including two NPRs duplicated in the IR regions. Among the six excluding the duplicated ones, there was found no correspondence to any nucleotide sequences reported to date (Appendix A). As mentioned in the previous section, an additional 1.15% to 25% of mitogenomic regions in certain plants is conjectured to be derived from nuclear genome sequences, although, at present, this cannot be verified owing to the current deficiency in sequence data. Thus, subject to further confirmation, we speculated that unidentified sequences in the *V. sambucifolia* f. *dageletiana* plastome are nuclear derived and have been incorporated into the plastome in the same manner whereby other nuclear DNA sequences have integrated into plastid genomes. However, it remains conceivable that these unidentified sequences could have originated directly from bacteria or fungi. Indeed, in this regard, it has been established that the plastomes of *Mankyua chejuense*, an early diverging species of eusporangiate fern, contain an approximate 8 kb insertion between *rps4* and *trnL*-UAA and that certain open reading frames in this insertion are likely to have been acquired from distantly related taxa such as algae or bacteria [66]. Moreover, although these open reading frames have also been identified in other fern organelles [67,68], most of the intergenic spacers between open reading frames detected in *M. chejuense* do not match any sequence regions in current nucleotide databases. These findings thus tend to imply that the identity of horizontally transferred non-coding sequences would not be ascertained based on nucleotide database searches. Accordingly, it can be anticipated that the accumulation of larger amounts of sequence data, including those of bacteria, fungi, and the nuclear DNA of terrestrial plants, will, in the future, enable us to gain a more comprehensive insight into the origins of these unidentified NPRs.

Nevertheless, the source of the foreign DNA sequence NPR_7 detected in the *V. sambucifolia* f. *dageletiana* plastome remains something of a conundrum. Although sequences similar to NPR_7 have infrequently been identified in some lineages of bacteria and fungi, this sequence was found to correspond most closely to certain regions in the DNA of bony fishes. Generally, gene transfer is mediated via physical contact, notably through inter-compartmental gene transfer within the same cell or horizontal gene transfer between endosymbionts and hosts or two physically linked species. In contrast, given their spatial isolation, it is difficult to envisage any similar mode of contact between bony fishes and *V. sambucifolia* f. *dageletiana*. On the basis of the terrestrial plant plastome database and BLAST results, we assume that the NPR_7 sequence has recently undergone to be transferred into the *V. sambucifolia* plastome during the evolution and that the ancestor of bony fishes contained an NPR_7-like region in its nuclear genome. 

As mechanisms to account for the similarity of sequences identified in the *V. sambucifolia* f. *dageletiana* plastome and bony fish, we propose three conceivable scenarios. The first of these envisages vertical inheritance from common ancestors of eukaryotes and bacteria (Figure 7A). Although most of the BLAST-identified sequences corresponding to NPR_7 are derived from bony fishes, we found that the NPR_7 sequence also rarely showed a certain degree of match to the messenger RNAs and DNAs of mosquitos, primates, apicomplexans, crustaceans, bivalves, and bacteria. It is thus conceivable that regions of NPR_7-like ancient DNA did not need to be maintained for survival. Accordingly, they have been less influenced by purifying selection during evolution and thus more readily degraded or deleted than coding sequences and sequence motifs. Moreover, whereas some of these non-essential DNA sequences harbored in the nuclear genomes of eukaryotes may have disappeared within certain lineages, others may have been retained, as in *V. sambucifolia* f. *dageletiana*. In the final event in this scenario, it is probable that the NPR_7 sequence underwent translocation from the nucleus to the plastid in a manner similar to that of other plastid components derived from nuclear DNA. 

In the second scenario, we posit that the NPR_7 region of *V. sambucifolia* f. *dageletiana* and NPR_7-like regions of bony fish originated from other organisms, such as bacteria (Figure 7B). The horizontal transfer of genes from bacteria to plants [69,70] and animals [71,72,73] is well documented, and the acquisition of genes from bacteria has similarly been detected in protists [74,75] and fungi [76]. Accordingly, the transmission of bacterial DNA into eukaryotes is probably a universal phenomenon. Consequently, the appearance of NPR_7-like sequences in diverse eukaryote lineages might be the result of independent horizontal gene transfer from bacteria to different lineages. 

In the third scenario, we speculate that NPR_7 was transferred from bony fishes to an ancestor of *V. sambucifolia* f. *dageletiana* via bacteria, fungi, or other vectors (Figure 7C). Although the transfer of genes between animals and terrestrial plants is considered to be an extremely rare evolutionary phenomenon, its likelihood should not be completely dismissed. Repetitive DNA sequences, referred to as transposable elements or transposons, have a tendency to relocate within genomes [77] and even surmount species boundaries [78,79]. For example, Lin et al. [80] revealed that the prominent *Penelope*-like elements, a group of arthropod retrotransposons, were transferred to a common ancestor of conifers, either directly or via vectors such as bacteria and fungi. 

Nevertheless, despite the plausibility of these three scenarios as explanations for the origin of the anomalous NPR_7 sequence, we still require more persuasive experimental evidence to fully comprehend this finding. For example, given the current limitations regarding database resources, we have been unable to establish whether NPR_7-like sequences are found in the genomes of a wider range of terrestrial plants or whether, with the exception of certain taxa such as *V. sambucifolia* f. *dageletiana*, most of these sequences have been eliminated during the course of evolution, as envisaged in the vertical inheritance scenario (Figure 7A). Furthermore, if NPR_7-like sequences are of bacterial origin, it is worth exploring as to why these sequences were identified in the genomes of only a few bacteria (Figure 7B) or why, if the sequence was derived from bony fishes via vector mediation, horizontal gene transfer has seemingly occurred only in an endemic plant isolated on a small island in the nearly 250 million years ago since the common ancestor of bony fishes diverged [81] (Figure 7C). Although, given the lack of relevant data, we are currently unable to resolve these questions with any degree of clarity, a maximum-parsimony scenario would tend to suggest that DNA derived from bony fish was transferred to *V. sambucifolia* f. *dageletiana* or its most recent common ancestor via intermediary vectors such as bacteria or fungi, whereas independent gene transfer from bony fish may have occurred in other, more distantly related taxa. 

### 3.3. Rapid Structural Mutation of Plastomes in the Genus Valeriana

With the exception of certain lineages, such as those of Campanulaceae [82], Geraniaceae [83,84], Oleaceae [85], Orchidaceae [86], and Orobanchaceae [26], the plastomes of flowering plants tend to be structurally conserved [17]. However, although the structures of plastomes in plants of the family Caprifoliaceae have appeared to be highly conserved and characterized by the elimination of *rpl2*, *rps19*, and *ycf1* from the typical IR [44], the recently sequenced plastomes of two *Dipsacus* [87] and a *Linnaeae* [88] species in this family have revealed retention of the typical IRs of angiosperms. In addition to contracted IR regions with respect to the *rpl2*, *rps19*, and *ycf1* loci, the boundaries of IR regions in the *V. sambucifolia* f. *dageletiana* plastome appear to have undergone a more complex series of shifts. In Caprifoliaceae, transcription of the *trnH* gene appears to be modified to a certain extent, owing to gene duplication in the LSC region throughout the family, including *V. officinalis* [44]. Contrastingly, in the *V. sambucifolia* f. *dageletiana* plastome, duplication of the *trnH* gene shows a different pattern, involving complete translocation into the IR region, similar to that seen in certain *Patrinia* species [43,44]. Furthermore, it is assumed that two expansions and one contraction of the IR have occurred in the plastomes of *V. sambucifolia* f. *dageletiana* and *V. officinalis,* at least in the IR/SSC border regions (Figure 2). Interestingly, we found that a majority of the NPRs detected in the *V. sambucifolia* f. *dageletiana* plastome are located in the vicinity of the *trnN*-GUU locus or within *ycf1*, which are typically close to each other at adjacent IR/SSC borders in angiosperms. In this regard, the mutation of transfer RNAs has been well documented. For example, the inversion of plastid DNA between distinct tRNA genes has been identified in cereals [89] and extensive rearrangements associated with tRNA genes in the plastome of *Trachelium caeruleum* have been reported [82]. Moreover, it has been found that tRNA genes in the plastomes of ferns appear to contribute to genome instability, leading to the insertion of mobile elements [68]. 

Although the association between IR/SSC border instability and the translocation of foreign DNA is ambiguous, given that IR/SSC borders in the plastomes of Caprifoliaceae plants tend to be highly conserved [44] and our finding that NPRs are generally located in the vicinity of tRNA genes, we assume that in the *V. sambucifolia* f. *dageletiana* plastome, foreign DNA has become inserted in sequences adjacent to the IR/SSC borders. Such insertion events are presumed to weaken the stability of IR/SSC borders, thereby providing conditions conducive to IR expansions/contractions. Consequently, we speculate that gene duplication, horizontal gene transfer, and IR expansions/contraction have occurred independently and recently in *Valeriana*, subsequent to the divergence of *V. sambucifolia* and *V. officinalis*. 

### 3.4. Genetic Consequence of Anagenetic Speciation on Ulleung Island

Among the species endemic to Ulleung Island, the plastome sequences of those reported to date have been found to be highly conserved with respect to both structure and gene content [90,91,92,93,94,95]. The plastome sequence of *V. sambucifolia* f. *dageletiana* reported herein thus appears to be a notable exception. Although Yang et al. [91] proposed the use of four highly variable regions to assess the genetic consequences of anagenetic speciation on Ulleung Island, these four loci are generally highly variable throughout angiosperms [23,96,97,98,99]. Therefore, there is currently no conclusive evidence supporting anagenetic speciation on Ulleung Island based on organelle genome evolution. 

In contrast to the plastome data obtained for other plant taxa distributed on Ulleung Island, we identified pronounced structural changes in the organellar genomes of *V. sambucifolia* f. *dageletiana*, including rearrangements and gene transfer. This, thus, raises the question as to why such structural variants have been detected only in the plastid genome of *V. sambucifolia* f. *dageletiana* and why the plastomes of 10 other species endemic to the island have a typical structure without any notable alterations. To resolve these issues, we recommend extensive comparative genomic analyses, focusing not only on plastomes but also on other genomes (mitogenomes and/or nuclear genomes) among island endemics and closely related mainland taxa.

## 4. Materials and Methods

### 4.1. Sampling, DNA Extraction, and Next-Generation Sequencing

Specimens of *V. sambucifolia* f. *dageletiana* (Figure 1) were collected from the plant’s natural habitat on Ulleung Island, Korea, and a voucher specimen has been deposited at Kyungpook National University. For the purposes of DNA extraction, we desiccated two leaves using silica gel and then ground these to a powder using a homogenizer. Total genomic DNA was extracted using a DNeasy Plant Mini Kit (Qiagen, Hilden, Germany) and was quantified using a NanoDrop spectrophotometer. Using this quantified DNA, three different libraries for next-generation sequencing (two paired-end libraries and one mate-pair library (5 kb paired distance)) were generated using the services of Macrogen, Seoul, Korea, and sequenced using the Illumina platform (Appendix A). Raw reads were trimmed using Trimmomatic 0.39 [100], with the options LEADING:10, TRAILING:10, SLIDINGWINDOWS:4:20, and MINLEN:50.

### 4.2. Assembly and Annotation of the Plastome and Mitogenome

The plastome of *V. sambucifolia* f. *dageletiana* was assembled and annotated using the protocol described by Kim and Chase [101], with the exception of read trimming. For the mitogenome, we used slightly modified versions of previously developed baiting and iteration strategies [16,102]. Having trimmed the raw reads, these were initially assembled using the de novo assembly programs Abyss [103,104] and Megahit [105,106] (Appendix A). Plastome contigs and contigs less than 10 kb in size were then filtered out, and contigs including mitochondrial fragments were extracted using BLASTn [107]. Thereafter, the reads were mapped to the contigs using the Geneious assembler [108] with zero mismatches and zero gaps for 25 iterations. The consensus contigs thus generated were aligned using de novo assembly, and the resultant contigs were repeatedly re-used as references until all contigs were incorporated into a single circular contig. 

### 4.3. Analysis of Repetitive Sequences in Organellar Genomes

Simple sequence repeats (SSRs) detected in both organellar genomes were identified using MISA-web, with the options 7, 5, 4, 3, 3, and 3 for mono- to hexa-nucleotides, respectively. In addition, dispersed repeats in the mitogenome were identified using BLASTn [107], with a word size of 7 and an expected value (e-value) of 1 × 10^−6^. To minimize overestimation of repeat values, BLAST hits with identical query start and end positions were removed. 

### 4.4. Identification of Mitochondrial Plastid DNAs in the Mitogenome of V. sambucifolia f. dageletiana

To investigate the occurrence MTPTs in the mitogenome of *V. sambucifolia* f. *dageletiana*, we blasted the mitogenome against 9025 plastome sequences of terrestrial plant origin (downloaded from the NCBI database) using BLASTn with an e-value of 1e^-5^ and a word size of 11. When queries corresponding to sequences in the database overlapped on the mitogenome, they were annotated using a single MTPT on the mitogenome of *V. sambucifolia* f. *dageletiana*. In addition, using BLASTn with an e-value of 1e^-5^ and a word size of 11, the mitogenome was blasted against 364 terrestrial plant mitogenome sequences downloaded from the NCBI database to detect non-mitogenomic regions. The different e-value thresholds applied to the aforementioned BLASTn searches were based on different total database lengths, which directly influenced the e-value. 

### 4.5. Identification of non-plastome-originated DNAs in the plastome of V. sambucifolia f. dageletiana

The plastome of *V. sambucifolia* f. *dageletiana* was blasted against Viridiplantae plastid DNAs downloaded from GenBank (1,419,429 sequences) using BLASTn with an e-value of 1e^−3^ and a word size of 11. The intervening sequences that showed no match with those in the database were extracted and to identify their origin were blasted against the NCBI nucleotide database (https://blast.ncbi.nlm.nih.gov/Blast.cgi accessed on 5 April 2021) using BLASTn with an e-value of 1e^−3^ and a word size of 11. 

## Figures and Tables

**Figure 1 ijms-22-03770-f001:**
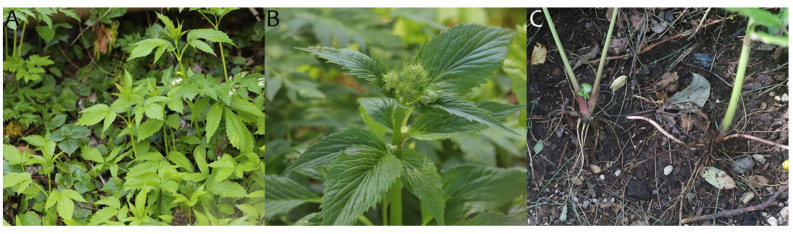
*Valeriana sambucifolia* f. *dageletiana* in its natural habitat on Ulleung Island, Korea. (**A**) General feature of entire plant; (**B**) Leaves and flower buds., (**C**) The base of stems and runners.

**Figure 2 ijms-22-03770-f002:**
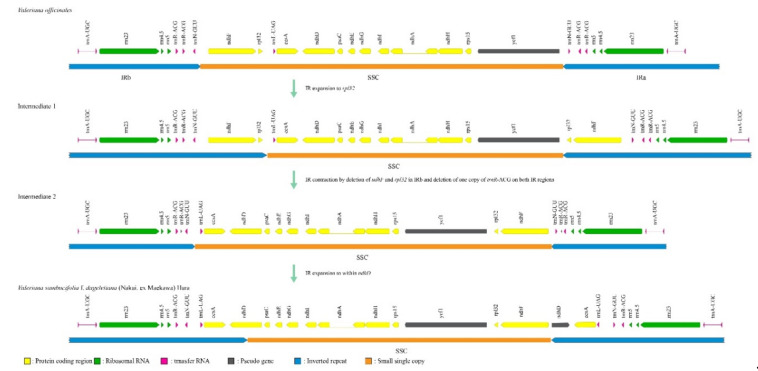
Comparison of inverted repeat (IR) expansion and contraction at the boundaries between IR and single-copy (SC) regions in the plastomes of *Valeriana sambucifolia* f. *dageletiana* and *Valeriana officinalis*.

**Figure 3 ijms-22-03770-f003:**
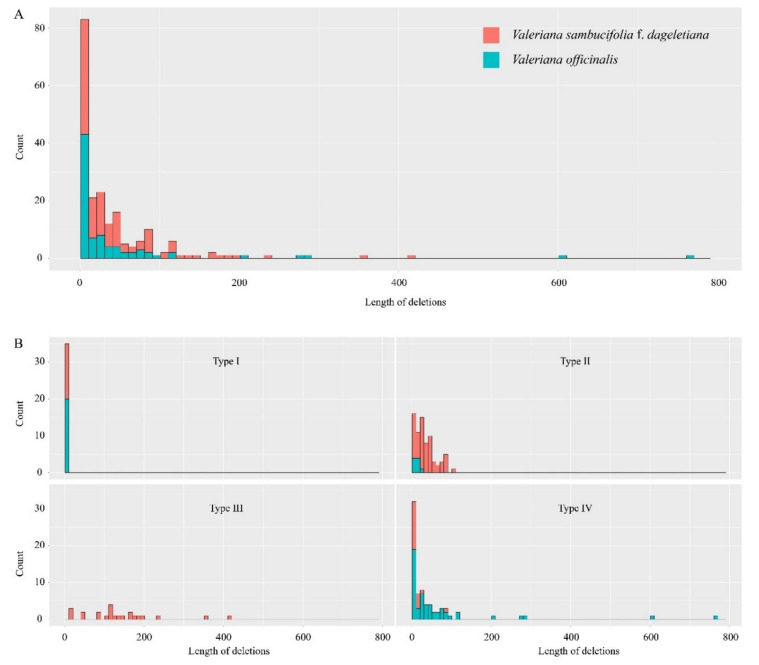
Deletions detected in the plastomes of *Valeriana sambucifolia* f. *dageletiana* (red bars) and *Valeriana officinalis* (green bars). (**A**) Total deletions. (**B**) The different types of deletions.

**Figure 4 ijms-22-03770-f004:**
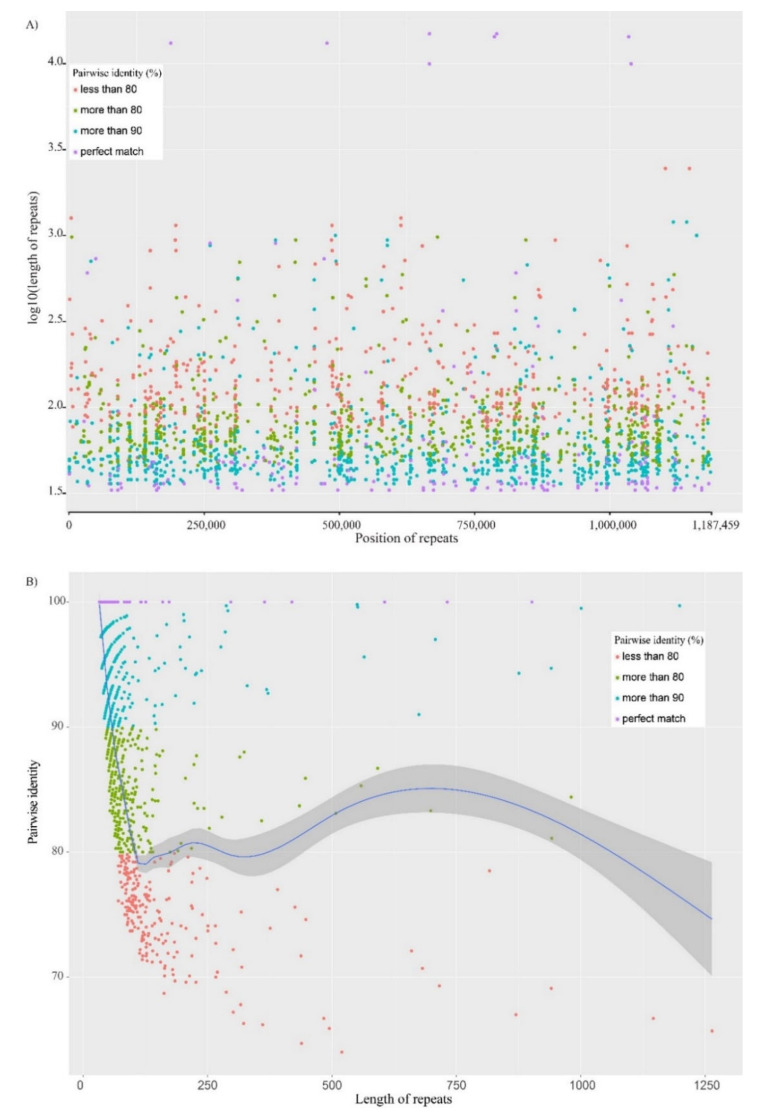
Dispersed repeats in the mitogenome of *Valeriana sambucifolia* f. *dageletiana*. Different-colored dots denote different percentage pairwise identities between dispersed repeats. (**A**) The length of repeats based on position. (**B**) Correlation between pairwise identity and the length of repeats.

**Figure 5 ijms-22-03770-f005:**
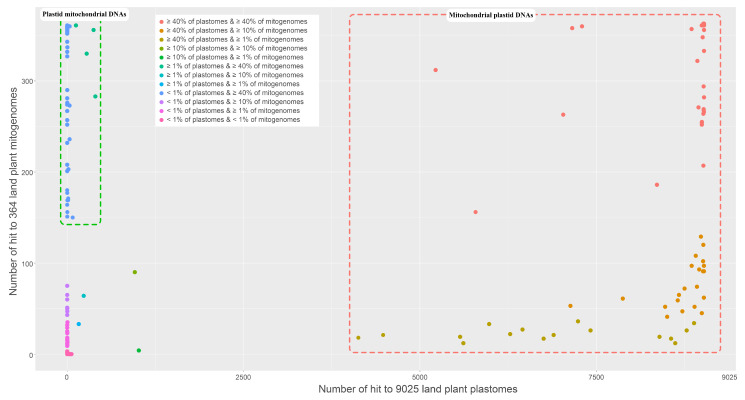
BLAST results for searches against 9025 plastomes (X axis) and 364 mitogenomes (Y axis) of terrestrial plants. Different-colored dots denote the difference in the percentage matches for subjects.

**Figure 6 ijms-22-03770-f006:**
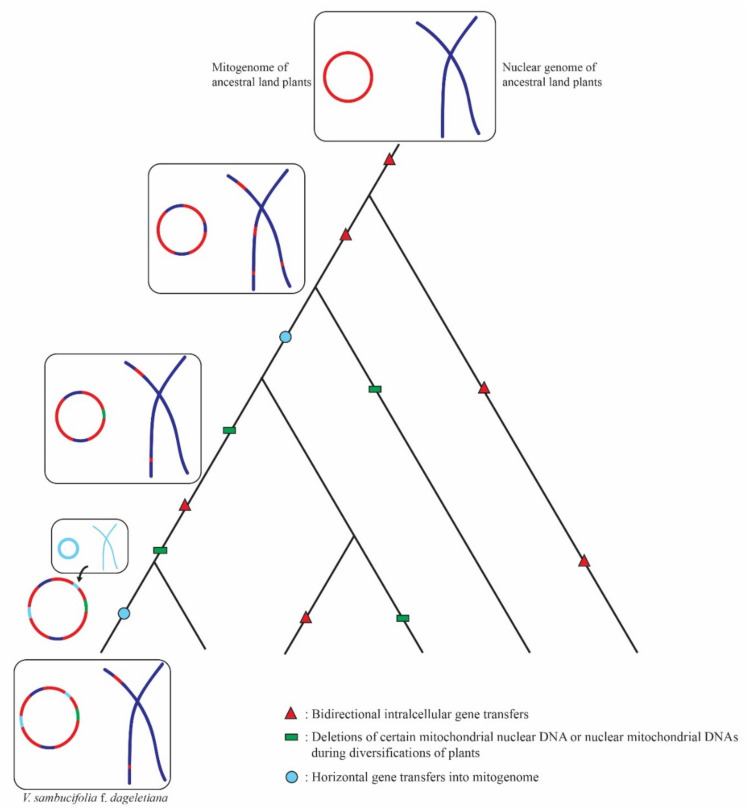
Model of the proposed origins of unidentified mitogenome sequences found in *Valeriana sambucifolia* f. *dageletiana*.

**Figure 7 ijms-22-03770-f007:**
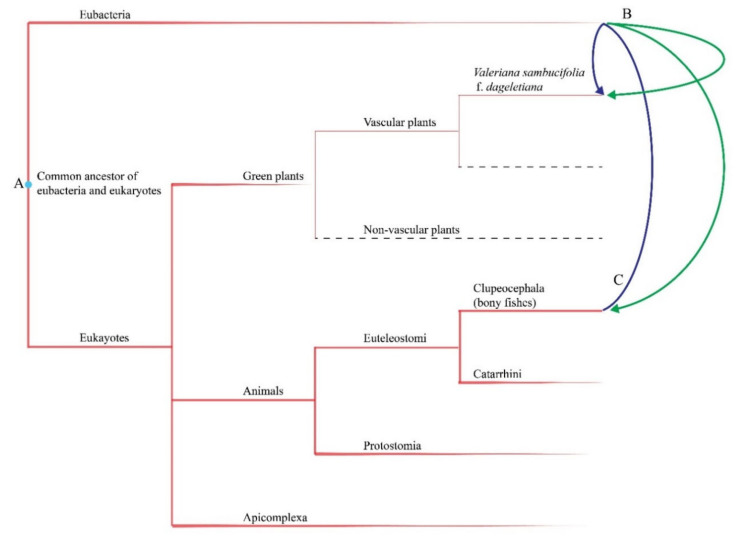
Three scenarios proposed to explain the occurrence of sequence region similarity between the plastome of *Valeriana sambucifolia* f. *dageletiana* and the nuclear genome of bony fishes. (**A**) Vertical inheritance from a common ancestor of eukaryotes and bacteria. (**B**) Independent gene transfer from bacteria. (**C**) Horizontal gene transfer from bony fishes to *V. sambucifolia* f. *dageletiana* via bacteria.

**Table 1 ijms-22-03770-t001:** Mitogenome-coding genes in *V. sambucifolia* f. *dageletiana* compared with those in species in the order Asterales. The shaded rows indicate conserved genes in the orders Asterles and Dipsacales.

Gene	Asterales															Dipsacales
	*Chrysanthemum boreale*	*Chrysanthemum indicum*	*Codonopsis lanceolata*	*Diplostephium hartwegii*	*Helianthus annuus 1*	*Helianthus annuus 2*	*Helianthus annuus 3*	*Helianthus annuus 4*	*Helianthus annuus 5*	*Helianthus annuus 6*	*Lactuca saligna*	*Lactuca sativa*	*Lactuca serriola*	*Paraprenanthes iversifolia*	*Platycodon grandiflorus*	*Valeriana sambucifolia* f. *dageletiana*
*atp1*	+ ^a^	+	+	+	+	+	+	+	+	+	+	+	+	+	+	+
*atp4*	+	+	+	+	+	+	+	+	+	+	+	+	+	+	+	+
*atp6*	+	+	+	+	+	+	D ^b^	+	+	+	+	+	+	+	+	+
*atp8*	+	+	+	+	+	+	+	+	+	+	+	+	+	+	+	+
*atp9*	+	+	+	+	+	+	+	+	+	+	+	+	+	+	+	+
*ccmB*	+	+	+	+	+	+	+	+	+	+	D	D	D	+	+	+
*ccmC*	+	+	+	+	+	+	+	+	+	+	+	+	+	+	+	+
*ccmFc*	+	+	+	+	+	+	+	+	+	+	+	+	+	+	+	+
*ccmFn*	+	+	+	+	+	+	+	+	+	+	+	+	+	+	+	+
*cob*	+	+	+	+	+	+	+	+	+	+	+	+	+	+	+	+
*cox1*	+	+	+	+	+	+	+	+	+	+	+	+	+	+	+	+
*cox2*	+	+	+	+	+	+	+	+	+	+	+	+	+	+	+	+
*cox3*	+	+	+	+	+	+	+	+	+	+	+	+	+	+	+	+
*matR*	+	+	+	+	+	+	+	+	+	+	+	+	+	+	+	+
*mttB*	+	+	+	+	+	+	+	+	+	+	+	+	+	+	+	+
*nad1*	+	+	+	+	+	+	+	+	+	+	+	+	+	+	+	+
*nad2*	+	+	+	+	+	+	+	+	+	+	+	+	+	+	+	+
*nad3*	+	+	+	+	+	+	+	+	+	+	+	+	+	+	+	+
*nad4*	+	+	+	+	+	+	+	+	+	+	+	+	+	+	+	+
*nad4L*	+	D	+	+	+	+	+	+	+	+	+	+	+	+	+	+
*nad5*	+	+	+	+	+	+	+	+	+	+	D	D	D	D	P ^c^	+
*nad6*	+	+	P	+	+	+	+	+	+	+	+	+	+	+	+	+
*nad7*	+	+	+	+	+	+	+	+	+	+	+	+	+	+	+	+
*nad9*	+	+	+	+	+	+	+	+	+	+	+	+	+	+	+	+
*rpl5*	+	+	- ^d^	+	+	+	+	+	+	+	+	+	+	+	-	+
*rpl10*	+	+	+	+	+	+	+	+	+	+	D	D	D	+	+	+
*rpl16*	P	P	+	+	+	+	+	+	+	+	+	+	+	+	+	P
*rps1*	P	+	+	+	-	-	-	-	-	-	-	-	-	P	+	P
*rps3*	+	+	+	+	+	+	+	+	+	+	+	+	+	+	+	+
*rps4*	+	+	+	+	+	+	+	+	+	+	+	+	+	+	+	+
*rps7*	-	-	+	-	-	-	-	-	-	-	-	-	-	-	+	+
*rps10*	-	-	-	-	-	-	-	-	-	-	-	-	-	-	-	+
*rps11*	-	P	-	P	+	+	+	+	+	+	P	P	P	P	-	-
*rps12*	+	+	+	+	+	+	+	+	+	+	+	+	+	+	+	+
*rps13*	+	+	+	+	+	+	+	+	+	+	+	+	+	+	+	P
*rps14*	P	P	-	P	P	P	P	P	P	P	P	P	P	P	-	+
*rps19*	-	-	-	-	-	-	-	-	-	-	-	-	-	-	-	P
*shd3*	-	-	P	-	-	-	-	-	-	-	-	-	-	-	+	-
*shd4*	+	+	-	+	P	P	P	+	P	P	P	P	P	P	+	-

^a^ +: present; ^b^ D: duplicated; ^c^ P: pseudogene or partial gene; ^d^ -: absent.

**Table 2 ijms-22-03770-t002:** The results of BLASTn searches for seven mitogenomes with or without whole-genome sequences.

Taxa	Accession of Mitogenome/Length (bp)	Accession of WGS ^a^	% of Non-Hit Regions	
			nt ^b^	nt + WGS
*Valeriana sambucifolia* f. *dageletiana*	Present study/1,187,459	-	31.7	-
*Platycodon grandiflorus*	NC_035958/1249,593	-	63.5	-
*Pinus taeda*	NC_039746/1,191,054	-	43.3	-
*Schisandra sphenanthera*	NC_042758/1,101,768	-	19.9	-
*Cucurbita pepo* subsp. *pepo*	NC_014050/982,833	NC_036638.1~NC_036657.1	46.2	21.2
*Mangifera indica*	CM021857/871,458	CM021837.1~CM021856.1	19.8	15.6
*Acer yangbiense*	CM017774/803,281	CM017761.1~CM017773.1	12.0	10.9

^a^ WGS: whole-genome sequences; ^b^ nt: nucleotide database.

**Table 3 ijms-22-03770-t003:** Number of organelle genomes against 149 regions in the mitogenome of *V. sambucifolium* f. *dageletiana* that corresponded to sequences in the plastomes of terrestrial plants.

Number of Queries	Number of Subjects Against Queries		Gene Transfer
	Percentage of 9025 Plastomes	Percentage of 364 Mitogenomes	
31	≥40	≥40	Plastome → Mitogenome (mitochondrial plastid DNA (MTPT))
21	≥40	<40 and ≥10	
16	≥40	<10 and ≥1	
1	<40 and ≥10	<40 and ≥10	
1	<40 and ≥10	<10 and ≥1	
4	<10 and ≥1	≥40	Mitogenome → Plastome (plastid mitochondrial DNA (PTMT))
1	<10 and ≥1	<40 and ≥10	
1	<10 and ≥1	<10 and ≥1	
42	<1	≥40	Mitogenome → Plastome (PTMT)
7	<1	<40 and ≥10	
17	<1	<10 and ≥1	
7	<1	<1	Nuclear genome → Organelle genomes

## Data Availability

Data are available from the Dryad Digital Repository, https://datadryad.org/stash/share/WB4rygpAm0IuoBDTuIgcMWaoH7I03UNPGy8jvIMKTWc (accessed on 5 April 2021).

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
