# Peer review of "Structural Mutations in the Organellar Genomes of Valeriana sambucifolia f. dageletiana (Nakai. ex Maekawa) Hara Show Dynamic Gene Transfer"

_ijms, 2021, doi:10.3390/ijms22073770_

Round 1
Reviewer 1 Report
This is a review of the manuscript “Structural mutations in two organellar genomes of Valeriana sambucifolia f. dageletiana (Nakai ex Maekawa) Hara show dynamic gene transfers during island speciation by the founder-flush effect” by H. T. Kim and J. S. Kim submitted for possible publication in the International Journal of Molecular Sciences.
Ullung Island is a very interesting situation, because nearly all (or perhaps all) of the endemic species have speciated anagenetically. That is, none has speciated cladogenetically to produce a complex of adaptively radiated taxa. This is not surprising due to the relatively uniform environment of the island and its youthful age (1.8 Ma). Because of this circumstance, the island is an excellent natural laboratory for studying speciation via transformation, and in fact, it may be the best place in the world for these sorts of investigations. A number of previous studies have focused on phylogenetic relationships of the island endemics with close relatives, mostly from the Korean peninsula and Japan. Other studies have examined patterns of genetic variation within and among populations of selected endemic species.
In my view, this manuscript is very interesting and should be publishable for the descriptive data relative to the organellar genomes within Valeriana, but two points seem to me problematic. The first is trying to explain the origin of the inexplicable NPR that apparently matches to a bony fish. Now, it is completely possible that some bacteria-mediated event did result in a HGT into the plastome of V. sambucifolia in the island, but I am suspicious that this could be the only explanation. Without more investigation into this situation, I would suggest that it be deleted from the paper. It is just too speculative as it stands.
The second problem derives from placing the results in context of founder-flush speciation. This concept, now 50 years old, has not been used much, if at all, with plant systems. To what extent it applies to anagenetic speciation is debatable, where genetic diversity accumulates over time within the uniform environment. This is another part that might better be deleted from the paper.
This may all relate to the selection of the title. I would recommend that the paper should focus entirely on the structural aspects and possible gene transfers and remove discussions regarding hypotheses of HGT and founder-flush speciation. A substitute title might simply end with “…dynamic gene transfers”, deleting “during island speciation by the founder-flush effect.” The paper should be descriptive, with these speculations kept to a minimum.
A number of laboratories are now successfully sequencing entire plastomes, but after the data are in hand, what is the significance? The data relate to understanding evolution of plastid sequences, especially regarding the nucleus and mitochondrion, but all of this is in context of molecular organellar evolution, not population genetics. It seems to me that this paper is trying hard to find significance of the data both phylogenetically and phylogeographically, whereas such connections are not easily seen. The paper does delve a bit into the question of genomic evolution within the land plants, but this is another broad avenue that this paper cannot deal with comprehensively. As a result, it comes across as another distraction from the good descriptive work.
If the manuscript is accepted, an editor needs to go over the English a bit to clean up some awkward sentences (e.g., line 54).
In summary, I recommend this paper be published but with a narrowing of the focus on the description of the organellar genomes in Valeriana.
Author Response
Dear Editor and reviewers,
Thank you for your kind advice for improving our manuscript. We agreed with your suggestion and revised the manuscript as follows, especially in comparative analysis of genome database.
According to the reviewer’s comments, we applied recent database of land plant genomes, e.g. plastid DNAs of viridiplantae for detecting NPRs and updating nucleotide collections, for our analysis.
We would like that this revised manuscript and our response will be acceptable to you.
Best regrds,
<Reviewer 1>
This is a review of the manuscript “Structural mutations in two organellar genomes of Valeriana sambucifolia f. dageletiana (Nakai ex Maekawa) Hara show dynamic gene transfers during island speciation by the founder-flush effect” by H. T. Kim and J. S. Kim submitted for possible publication in the International Journal of Molecular Sciences.
Ullung Island is a very interesting situation, because nearly all (or perhaps all) of the endemic species have speciated anagenetically. That is, none has speciated cladogenetically to produce a complex of adaptively radiated taxa. This is not surprising due to the relatively uniform environment of the island and its youthful age (1.8 Ma). Because of this circumstance, the island is an excellent natural laboratory for studying speciation via transformation, and in fact, it may be the best place in the world for these sorts of investigations. A number of previous studies have focused on phylogenetic relationships of the island endemics with close relatives, mostly from the Korean peninsula and Japan. Other studies have examined patterns of genetic variation within and among populations of selected endemic species.
In my view, this manuscript is very interesting and should be publishable for the descriptive data relative to the organellar genomes within Valeriana, but two points seem to me problematic. The first is trying to explain the origin of the inexplicable NPR that apparently matches to a bony fish. Now, it is completely possible that some bacteria-mediated event did result in a HGT into the plastome of V. sambucifolia in the island, but I am suspicious that this could be the only explanation. Without more investigation into this situation, I would suggest that it be deleted from the paper. It is just too speculative as it stands.
Response : V. sambucifolia f. dageletiana and bony fishes are spatially isolated from each other. So We thought that if there had been gene transfers between them, it might occur indirectly. We agree that our discussion about the origin of NPRs is speculative but we didn’t jump to conclusion. Based on the result in which some of NPRs were hit to bacterial DNAs, we suggested three possible scenarios but did not emphasize this in the manuscript.
The second problem derives from placing the results in context of founder-flush speciation. This concept, now 50 years old, has not been used much, if at all, with plant systems. To what extent it applies to anagenetic speciation is debatable, where genetic diversity accumulates over time within the uniform environment. This is another part that might better be deleted from the paper.
Response : We removed the context of founder-flush speciation from the paper.
This may all relate to the selection of the title. I would recommend that the paper should focus entirely on the structural aspects and possible gene transfers and remove discussions regarding hypotheses of HGT and founder-flush speciation. A substitute title might simply end with “…dynamic gene transfers”, deleting “during island speciation by the founder-flush effect.” The paper should be descriptive, with these speculations kept to a minimum.
Response : We changed title and deleted context of founder-flush speciation from the paper. However, without the hypotheses of HGT, NPRs do not mean anything. At least, we have to debate how foreign DNAs similar to bony fish’s ones detected in the plastome of V. sambucifolia f. dageletiana. So we just simply suggested the senarios and tried not to over-discuss about them.
A number of laboratories are now successfully sequencing entire plastomes, but after the data are in hand, what is the significance? The data relate to understanding evolution of plastid sequences, especially regarding the nucleus and mitochondrion, but all of this is in context of molecular organellar evolution, not population genetics. It seems to me that this paper is trying hard to find significance of the data both phylogenetically and phylogeographically, whereas such connections are not easily seen. The paper does delve a bit into the question of genomic evolution within the land plants, but this is another broad avenue that this paper cannot deal with comprehensively. As a result, it comes across as another distraction from the good descriptive work.
If the manuscript is accepted, an editor needs to go over the English a bit to clean up some awkward sentences (e.g., line 54).
Response : After revising the manuscript, we had been serviced again for editing the English overall.
In summary, I recommend this paper be published but with a narrowing of the focus on the description of the organellar genomes in Valeriana.
Response : We revised the manuscript to simply focus on the features of two organellar genomes as we can.
Reviewer 2 Report
In the submitted manuscript, the authors sequenced and assembled both plastid and mitochondrial genomes of Valeriana sambucifolia f. dageletiana endemic to an island in Korea. Using BLAST searches they identified non-plastome originated sequences in V. sambucifolia f. dageletiana’s plastome. Using the same approach, they also showed sequenced that could have been transferred from plastome to mitochondrial genome in Valeriana and vice versa. I found the results were interesting and would be of interest of IJMS readers. However, the validation of the results needs to be strengthened. Below are my comments
- Identification of Non-plastome regions (NPRs). According to the authors, NPRs were identified as the plastome regions in V. sambucifolia that did not hit any plastomes of land plants. I found two issues with this approach (1) It’s not clear why the authors exclude the plastomes from non-land plants, such as algae. I would suggest the authors expand their search by including algal plastomes. (2) The NPR sequences or genomic coordinates should be provided in the manuscript so that reviewers/readers can examine the results.
- Potential source of NPRs. To identify the putative source of NPRS, the authors did Blast searches using NPRs as query against nucleotide collection database (nr). Although these blast search yielded hits, which were claimed as the putative source of these NPRs, no reciprocal blast was conducted to verify these results. I would urge the authors to include reciprocal blast to verify the origin of NPRs.
- The identification of MTPTs and PTMTs. In table 2, the manuscript reports the percentage of plastomes and mitogenomes in the database were Blast hit by V. sambucifolium’s mitogenome and uses three cutoffs, 40%, 10%, 1% as threshold to identify MTPTs and PTMTs. I found this is problematic. (1) the rational of using these three cutoffs were not provided or justified. (2) Without a phylogenetic context, the direction of intergenomic gene transfer cannot be established. Please clarify on this.
- Founder-Flush Speciation Theory. The citation [101] on this theory is incomplete. The link between this theory and the organellar genomic structure diversity in the target species presented in this manuscript is purely speculative. Without population-level data from multiple individual, it’s not possible to verify or falsify whether there is a link between founder-flush speciation theory and genomic changes. I would suggest the author revisit their conclusion and modify the title.
- Figure 4(b). The correlation between repeat length and pairwise identify. It’s not clear what the grey area means in the figure. It seems that the authors used the “smooth” function in ggplot, but not a linear correlation. Please clarify.
Author Response
Dear Editor and reviewers,
Thank you for your kind advice for improving our manuscript. We agreed with your suggestion and revised the manuscript as follows, especially in comparative analysis of genome database.
According to the reviewer’s comments, we applied recent database of land plant genomes, e.g. plastid DNAs of viridiplantae for detecting NPRs and updating nucleotide collections, for our analysis.
We would like that this revised manuscript and our response will be acceptable to you.
Best regards,
<Reviewer 2>
In the submitted manuscript, the authors sequenced and assembled both plastid and mitochondrial genomes of Valeriana sambucifolia f. dageletiana endemic to an island in Korea. Using BLAST searches they identified non-plastome originated sequences in V. sambucifolia f. dageletiana’s plastome. Using the same approach, they also showed sequenced that could have been transferred from plastome to mitochondrial genome in Valeriana and vice versa. I found the results were interesting and would be of interest of IJMS readers. However, the validation of the results needs to be strengthened. Below are my comments
- Identification of Non-plastome regions (NPRs). According to the authors, NPRs were identified as the plastome regions in V. sambucifolia that did not hit any plastomes of land plants. I found two issues with this approach (1) It’s not clear why the authors exclude the plastomes from non-land plants, such as algae. I would suggest the authors expand their search by including algal plastomes. (2) The NPR sequences or genomic coordinates should be provided in the manuscript so that reviewers/readers can examine the results.
Response : We changed the database for detecting NPRs to viridiplantae plastid DNAs deposited in GenBank (1,419,429 sequences) and provided the information of sequences and positions in detail.
- Potential source of NPRs. To identify the putative source of NPRS, the authors did Blast searches using NPRs as query against nucleotide collection database (nr). Although these blast search yielded hits, which were claimed as the putative source of these NPRs, no reciprocal blast was conducted to verify these results. I would urge the authors to include reciprocal blast to verify the origin of NPRs.
Response : When we interpret blast results, an expect value is one of the main informative values. This e-value is affected by the database size. Therefore, we did blast search of Valeriana plastome against viridiplantae plastid DNAs not vice versa. Because, if we did blast search of viridiplantae plastid DNAs against Valeriana plastome, the database size reduces dramatically and an e-value also reduces as compared with the previous one. That is why we don’t think that it has to do a reciprocal blast search. It is usually used for detecting orthologous genes between small numbers of taxa.
- The identification of MTPTs and PTMTs. In table 2, the manuscript reports the percentage of plastomes and mitogenomes in the database were Blast hit by V. sambucifolium’s mitogenome and uses three cutoffs, 40%, 10%, 1% as threshold to identify MTPTs and PTMTs. I found this is problematic. (1) the rational of using these three cutoffs were not provided or justified. (2) Without a phylogenetic context, the direction of intergenomic gene transfer cannot be established. Please clarify on this.
Response : 1) We used three thresholds of hit organellar genomes percentage against the mitogenome regions of V. sambucifolia f. dageletiana for recognizing the direction of gene transfers between intercompartments based on how data were categorized by the minimum thresholds. 2) The chloroplast genomes have generally been conserved for long time except certain lineages like heterotrophs.
- Founder-Flush Speciation Theory. The citation [101] on this theory is incomplete. The link between this theory and the organellar genomic structure diversity in the target species presented in this manuscript is purely speculative. Without population-level data from multiple individual, it’s not possible to verify or falsify whether there is a link between founder-flush speciation theory and genomic changes. I would suggest the author revisit their conclusion and modify the title.
Response : We removed the mentions about founder-flush speciation throughout the manuscript.
- Figure 4(b). The correlation between repeat length and pairwise identify. It’s not clear what the grey area means in the figure. It seems that the authors used the “smooth” function in ggplot, but not a linear correlation. Please clarify.
Response : We revised the sentence.